# Effect of Foot-and-Mouth Disease Vaccination on Acute Phase Immune Response and Anovulation in Hanwoo (*Bos taurus coreanae*)

**DOI:** 10.3390/vaccines9050419

**Published:** 2021-04-22

**Authors:** Daehyun Kim, Joonho Moon, Jaejung Ha, Doyoon Kim, Junkoo Yi

**Affiliations:** 1Livestock Research Institute, 186 Daeryongsan-ro, Anjeong-myeon, Yeongju 36052, Gyeongsangbuk-do, Korea; chunja2411@korea.kr (D.K.); hjjggo@korea.kr (J.H.); kdy51311@naver.com (D.K.); 2Lartbio Co., Ltd., 12th Floor, 234 Teheran-ro, Gangnam-gu, Seoul 06221, Korea; joonhomoon@lartbio.com

**Keywords:** FMD vaccine, anovulation, acute phase immune response, artificial insemination, Hanwoo

## Abstract

Vaccination against foot-and-mouth disease is the most common method for preventing the spread of the disease; the negative effects include miscarriage, early embryo death, lower milk production, and decreased growth of fattening cattle. Therefore, in this study, we analyze the side effects of vaccination by determining the acute immune response and ovulation rate after vaccinating cows for foot-and-mouth disease. The test axis was synchronized with ovulation using 100 Hanwoo (*Bos taurus coreanae*) cows from the Gyeongsangbuk-do Livestock Research Institute; only individuals with estrus confirmed by ovarian ultrasound were used for the test. All test axes were artificially inseminated 21 days after the previous estrus date. The control group was administered 0.9% normal saline, the negative control was injected intramuscularly with lipopolysaccharide (LPS; 0.5 µg/kg), and the test group was administered a foot-and-mouth disease virus vaccine (FMDV vaccine; bioaftogen, O and A serotypes, inactivated vaccine) 2, 9, and 16 days before artificial insemination. White blood cells and neutrophils increased significantly 1 day after vaccination, and body temperature in the rumen increased for 16 h after vaccination. Ovulation was detected 1 day after artificial fertilization by ovarian ultrasound. The ovulation rates were as follows: control 89%, LPS 60%, FMDV vaccine (−2 d) 50%, FMDV vaccine (−9 d) 75%, and FMDV vaccine (−16 d) 75%. In particular, the FMDV vaccine (−2 d) test group confirmed that ovulation was delayed for 4 days after artificial insemination. In addition, it was confirmed that it took 9 days after inoculation for the plasma contents of haptoglobin and serum amyloid A to recover to the normal range as the main acute immune response factors. The conception rate of the FMDV vaccine (−2 d) group was 20%, which was significantly lower than that of the other test groups.

## 1. Introduction

Foot-and-mouth disease (FMD) is a common disease in cattle (*Bos taurus*, *Bos indicus*) that leads to economic loss [1]. Therefore, the governments of many countries, including those in North America, Western Europe, South America, and Asia, have developed vaccination strategies to prevent the introduction and spread of foot-and-mouth disease virus (FMDV) in cattle [2]. FMDV vaccination is a powerful strategy to control the proliferation of FMDV [3]. Various FMDV vaccines, such as adeno and viral vector-based, virus-like particles, peptide and DNA vaccines, plant-based vaccines, and inactivated vaccines, have been used [4,5,6,7]. The most widely used vaccine in the world is the inactivated FMDV vaccine (O and A serotypes) [5,8].

However, negative effects of FMDV vaccination (early embryo loss, sperm infertility, increased acute-phase reaction protein levels, and decreased milk production) have been reported in previous studies [8,9,10]. Administration of FMDV vaccines 30 days after artificial insemination increases early pregnancy loss in dairy cows [8]. Increased rectal temperature and early pregnancy loss have been explained by the acute-phase immune reaction [8,11].

The mRNA levels of interleukin (IL)-1, IL-6, IL-8, and interferon-alpha increase in porcine T-cells infected with FMDV [12]. In addition, anovulation has been confirmed to be an inflammation model by directly injecting lipopolysaccharide (LPS) into ovarian follicles and by intravenous injection in cattle [13,14]. The expression of genes related to the acute-phase immune response, such as steroidogenic acute regulatory protein, Toll-like receptor 4, and tumor necrosis factor alpha (TNF- α), decreases significantly upon intravenous LPS injection [13].

FMDV-vaccine-induced inflammation increases inflammatory cytokines, such as TNF-α, IL-1β, and IL-8, by increasing acute-phase proteins such as haptoglobin and serum amyloid A (SAA) [11,12]. The increase in inflammatory cytokines in response to FMDV vaccines reduces 17β-estradiol (E2) and increases prostaglandin F2α (PGF2α) and body temperature [8,11,15]. Previous studies have reported that ovulation is delayed due to a relative decrease in progesterone (P4) and luteinizing hormone (LH) due to the imbalance between E2 and P4 [11,14].

The acute-phase reaction accompanied by inflammation has negative effects on pregnancy maintenance and conception rates [8,11]. Inflammation caused by FMDV vaccines increases the levels of acute-phase proteins and inflammatory cytokines and delays ovulation [3,11]. We hypothesize that administering the FMDV vaccine before artificial insemination would stimulate the acute-phase protein reaction and increase the anovulation rate.

Although it has been reported that FMDV vaccines increase early embryo loss in cattle after artificial insemination, no in vivo studies on the effects of FMDV vaccines on acute-phase protein reaction, anovulation, or the conception rate before artificial insemination have been reported in Korean cattle. Therefore, this study aims to investigate changes in acute-phase protein levels and ovulation rates when an FMDV vaccine is administered at different times before artificial insemination.

## 2. Materials and Methods

### 2.1. Animals

The animal experiment was approved by the Institutional Animal Care and Use Committee (#106) of the Gyeongsangbuk-do Livestock Research Institute, and all applicable national laws and policies regarding the care and use of animals were observed during the experiment. In total, 120 mature female Korean cattle (34.1 ± 1.45 months old, 1.1 ± 0.10 parity) were used. The heifers were housed in a stanchion barn with sufficient space during the experiment and were given feed according to the Korean feeding standard program. Rice straw, mineral blocks, and water were fed ad libitum.

### 2.2. Ruminal Temperature

Ruminal wireless sensors (LiveCare; ULikeKorea, Seoul, Korea) were inserted into 60 cows (Control, LPS, and FMDV vaccine groups; 20 cows each) to monitor ruminal temperature. This technique has been validated in a dairy cattle experiment with rumen-cannulated cows [16]. The sensor measured ruminal temperature every 10 min and transmitted the data in real-time to a base station using the LiveCare system. The sensors were 125 mm long, 36 mm in diameter, and weighed 200 g. This method of collecting ruminal body temperature was reported by Kim et al. [16,17].

### 2.3. Plasma Collection and Analysis of Complete Blood Count (CBC)

Blood samples were collected from all groups at −16 d to 4 d, according to an ovulation synchronization protocol (Figure 1). Briefly, EDTA-blood sample tubes were placed on a roller mixer and rolled at 33 rpm and 16 mm amplitude for proper mixing. White blood cells (WBCs; 1000 cell/µL), neutrophils (NEUs, 1000 cell/µL), eosinophils (EOSs, 1000 cell/µL), basophils (BASOs, 1000 cell/µL), monocytes (MONOs, 1000 cell/µL), and lymphocytes (LYMs, 1000 cell/µL) were analyzed with an automated hematology analyzer (ProCyte Dx Hematology Analyzer Vet; IDEXX Laboratories Inc., Westbrook, ME, USA).

### 2.4. Vaccination, Ovum Synchronization, and Ovulation and Pregnancy Tests

A combined protocol (Day 0 = day of artificial insemination) was performed to determine the normal cycle of the cows (Figure 1). The cows were synchronized by inserting an intravaginal drug-releasing device containing progesterone (CIDR, Cue-Mate, 1.56 g progesterone; Bioniche Animal Health, Belleville, ON, Canada) for 7 days. The cows were injected with 25 mg of a PGF2α analog (Lutalyse, dinoprost tromethamine 5 mg/mL; Zoetis, Fairfield, NJ, USA) 7 days after removing the CIDR. The cows were treated with 250 µg gonadotropin-releasing hormone (Gonadon, gonadorelin acetate 100 µg/mL; Dong Bang Co., Seoul, South Korea) 2 days later. We assessed the ovulation rate by rectal palpation and transrectal ultrasonography. As a result, 100 of the 120 experimental cows were confirmed to be normal cycling, while the other 20 cows had problems such as ovarian cyst, corpus luteal cyst, anovulation, or delayed ovulation. No cows with abnormal ovaries or uterus were used in this study.

The FMDV vaccine (−16 d) group received a 50% protective dose (PD_50_) intramuscular (i.m.) injection of the FMDV vaccine (Bioaftogen, FMDV vaccine O1 Campos, A24 Cruzeiro and A2001 Argentina; Biogénesis Bagó, Buenos Aires, Argentina) at −16 d (16 days before artificial insemination) after one estrous cycle. The FMDV vaccine (−9 d) group received a 2 mg i.m. FMDV vaccine injection at −9 d. The FMDV vaccine (−2 d) group received a 2 mg i.m. FMDV vaccine injection at −2 d. The negative control group was administered 2.5 mg/kg body weight LPS (*Escherichia coli* O55: B5; Sigma Chemical Co., St. Louis, MO, USA). The control group was administered 0.9% physiological saline (saline). Ovulation and pregnancy were determined 40 days after artificial insemination by transrectal ultrasonography (HS-101V; Honda, Tokyo, Japan) and reconfirmed after 100 and 200 days of pregnancy.

### 2.5. Concentrations of Haptoglobin and Serum Amyloid A (SAA)

Plasma haptoglobin and SAA levels were measured using an adapted colorimetric assay, a haptoglobin enzyme-linked immunosorbent assay (ELISA) kit (Catalogue No. E-10HPT; ICL Inc., Portland, OR, USA), and the SAA ELISA kit (Catalogue No. TP 802; Tridelta Development Ltd., Kildare, Ireland.). Absorbance was detected at 450 nm using an ELISA reader (Gen5; BioTek, Seoul, Korea).

### 2.6. Statistical Analysis

Statistical analysis was performed using GraphPad Prism (version: 8.0.1, GraphPad Software Inc., La Jolla, CA, USA). Differences in ruminal temperature, CBC, plasma haptoglobin, and SAA concentrations were analyzed by one-way analysis of variance. The conception rate was analyzed using the Mantel–Haenszel procedure in R (version 3.6.2, The R Foundation for Statistical Computing, Vienna, Austria). A *p*-value < 0.05 was considered to indicate significance.

## 3. Results

The differences in mean ruminal temperature of the control, LPS, and FMDV vaccine groups are shown in Figure 2. After administration, the hourly mean ruminal temperature in the FMDV vaccine group increased for more than 20 h, while the LPS group (i.m. injection of LPS, 2.5 mg/kg of body weight) increased slightly compared to the control group (i.m. injection of 0.9% physiological saline, 10 mL/head) over 10 h (*p* < 0.05). The decrease in ruminal temperature every 24 h indicates the times of feeding and water consumption.

As shown in Figure 3, the number of WBCs increased significantly in the LPS group compared to the FMDV vaccine group on Day 1 (*p* < 0.05). The number of WBCs of the FMDV vaccine and LPS groups was higher than the control group. WBCs returned to normal levels 4 days after administration in all groups. The percentages of leucocytes and NEUs increased in the LPS and FMDV vaccine groups 1 day after administration compared to those in the control group (Figure 4).

Delayed ovulation was observed in the LPS (−2 d) and FMDV vaccine (−2 d) groups 1 day after artificial insemination, with an ovulation rate of about 60%. The ovulation rate in the LPS (−2 d) and FMDV vaccine (−2 d) groups was 80% 4 days after artificial insemination (Figure 5).

The plasma levels of haptoglobin and SAA were greater in the LPS (−2 d) and FMDV vaccine (−2 d) groups 2 days after artificial insemination (= 4 days after administration) compared to the control group. The plasma level of haptoglobin increased in the FMDV vaccine (−2 d) group for 4 days after artificial insemination (= 6 days after administration). The plasma levels of haptoglobin were highest 4 days after administration in the FMDV vaccine (−16 d) group and 3 days after administration in the FMDV vaccine (−9 d) group. Serum amyloid A levels were highest 4 days after administration in the FMDV vaccine (−16 d) group and 3 days after administration in the FMDV vaccine (−9 d) group (Figure 6). The pregnancy rate in the FMDV vaccine (−2 d) group was significantly lower than that in the other groups (Table 1).

## 4. Discussion

FMDV vaccination is the best method to prevent FMD. The vaccine is injected twice every 6 months [18]. The livestock community has managed the proliferation of FMDV, and an inactivated FMDV vaccine is the most widely used method [5]. However, FMDV vaccines are accompanied by various negative effects, such as decreased reproductive performance and milk production, as well as increased embryo loss and acute-phase protein levels [8,9].

However, no in vivo study has been reported regarding the conception rate and acute-phase protein reaction after FMDV vaccine administration before artificial insemination in Korean cattle. The negative effects of FMDV vaccines on pregnancy loss in dairy cows have been reported in previous studies [8]. The plasma levels of haptoglobin and acute-phase proteins increase, and the pregnancy rate is lower than that of the control group in cows administered an FMDV vaccine 30 days after artificial insemination [8]. In the present study, we confirmed the lowest pregnancy rate after administration of the FMDV vaccine 2 days before artificial insemination.

Changes in body temperature in cows are highly correlated with physiological mechanisms, and wireless sensors have been used to manage breeding and reproductive performance as well as to detect disease [19,20]. Ferreira et al. reported that rectal temperature increases in cows after FMDV vaccine administration, and our ruminal temperature results were similar [8,21]. Body temperature rises more rapidly than usual within ~4 h after LPS injection in the direct intravenous injection model [14]. In addition, the imbalance in major ovulatory hormones, such as P4, E2, and LH, is accompanied by an acute-phase immune response to the FMDV vaccine [8,11,14].

We observed changes in acute-phase proteins such as haptoglobin and SAA. Previous studies have reported an increase in inflammatory cytokines and gene expression related to the acute-phase immune response in response to FMDV vaccines. Inactivated FMDV vaccine contains an oil-based adjuvant that induces the innate immune response, including inflammation and the acute-phase reaction that is associated with pregnancy loss [3,8,11,22]. The acute-phase immune response involves the synthesis of prostaglandins and acute-phase proteins, such as haptoglobin and SAA [11,22]. Increases in the levels of acute-phase proteins delay ovulation by increasing proinflammatory cytokines and endometrial PGF2α levels [11]. Injecting an FMDV vaccine 2 days before artificial insemination elicits an acute-phase response, as represented by the increased ruminal temperature and plasma haptoglobin and SAA levels in our results. Ferreira et al. also reported that FMDV vaccines increase plasma haptoglobin concentrations for up to 120 h; the plasma haptoglobin and SAA levels in the FMDV vaccine (−16 and −9 d) groups increased similarly [8,11].

We used the LPS-induced model to verify anovulation caused by inflammation. Delayed ovulation due to intrafollicular LPS has been reported in cows [13,23]. However, no study has reported anovulation in response to an i.m. injection of FMDV vaccine before artificial insemination in cattle. Our results indicate that the ovulation rates of the FMDV vaccine (−2 d) and LPS (−2 d) groups were lower than those of the other groups. Increases in acute-phase proteins (haptoglobin and SAA) and anovulation may also impair fertility and preimplantation processes. Therefore, FMDV-vaccine-induced inflammation increases acute-phase proteins, and ovulation is delayed due to increased gene expression associated with inflammatory cytokines and the acute-phase immune response.

## 5. Conclusions

In summary, injecting Korean cattle with an FMDV vaccine decreased the ovulation rate when administered 2 days before artificial insemination compared with 9 or 16 days before artificial insemination. Acute-phase reaction proteins, such as haptoglobin and SAA, were increased for 8–9 days after the FMDV vaccine was administered. The pregnancy rate decreased two-fold when the FMDV vaccine was injected 2 days before artificial insemination. These results suggest that FMDV vaccine protocols should be changed to minimize the negative effects related to reproductive performance in cows. Artificial insemination should only be considered 10 days after an FMDV vaccine is administered to avoid the acute-phase immune reaction and anovulation.

## Figures and Tables

**Figure 1 vaccines-09-00419-f001:**
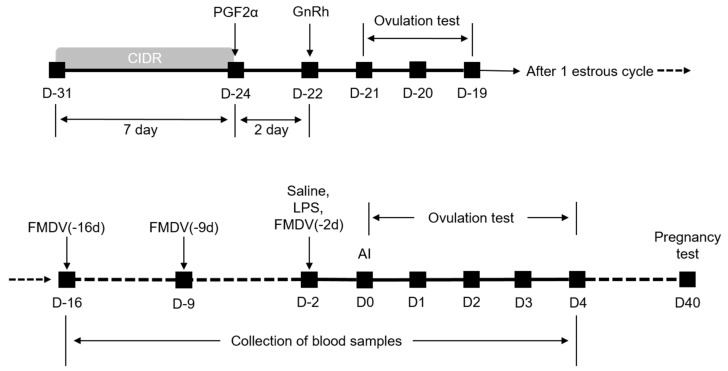
Schematic diagram of ovarian synchronization with the combined intravaginal progesterone releasing device (CIDR) used in this study. Normally, cycling cows are selected according to the results of an ovulation test by transrectal ultrasonography. All animals were artificially inseminated after one estrous cycle. Lipopolysaccharide (LPS; 2.5 mg/kg of body weight) or saline solution (control group) was injected i.m. 2 days before artificial insemination. Foot-and-mouth disease virus vaccine (FMDV vaccine; 2 mL/head) was injected i.m. 2, 9, or 16 days before artificial insemination. Transrectal ultrasonography was conducted 0–4 d after artificial insemination to determine ovulation. The pregnancy test was conducted 40 days after artificial insemination. Blood samples were collected in vacuum tubes containing EDTA from −16 to 4 d. D: day.

**Figure 2 vaccines-09-00419-f002:**
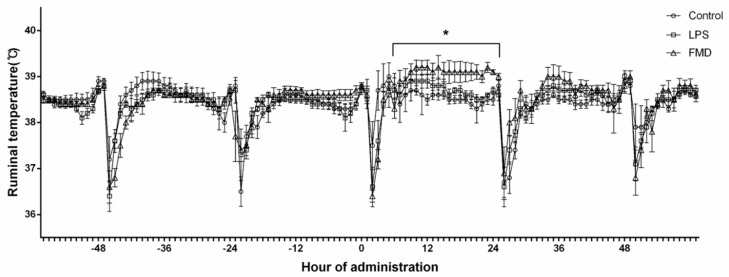
Ruminal temperature of the FMDV vaccine, LPS, and control groups before and after administration (*n* = 60). The black lines with symbols (○, □, △) represent the average 1-h ruminal temperature; 0 h is the time of administration. The control group (○, *n* = 20) was administered an i.m. injection of 0.9% physiological saline (10 mL/head), the LPS group (□, *n* = 20) was administered an i.m. injection of lipopolysaccharide (2.5 mg/kg of body weight), and the FMDV vaccine group (△, *n* = 20) was administered an i.m. injection of foot-and-mouth disease vaccine (6 PD_50_, 2 mL/head). * Experimental group comparisons within hours; *p* ≤ 0.05.

**Figure 3 vaccines-09-00419-f003:**
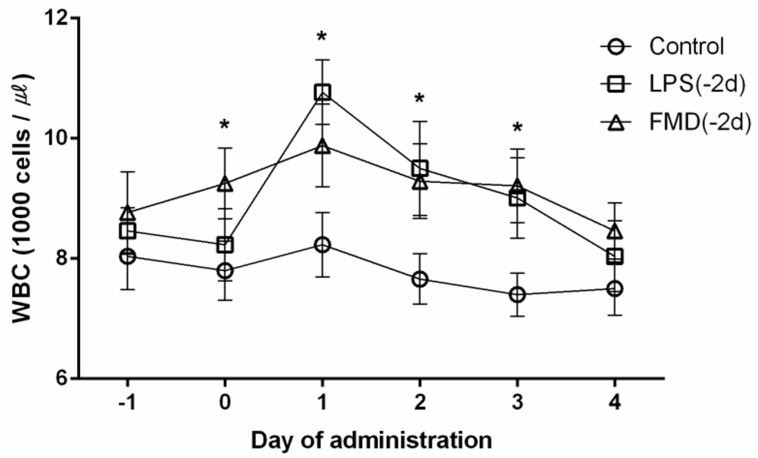
The white blood cell (WBC) concentrations in the different treatments (*n* = 60). The black lines with symbols (○, □, △) represent the WBC counts; 0 d is the time of administration. The control group (○, *n* = 20) was administered an i.m. injection of 0.9% physiological saline (10 mL/head), the LPS group (□, *n* = 20) was administered an i.m. injection of lipopolysaccharide (2.5 mg/kg of body weight), and the FMDV vaccine group (△, *n* = 20) was administered an i.m. injection of foot-and-mouth disease vaccine (6 PD_50_, 2 mL/head). * Experimental group comparisons within days; *p* < 0.05.

**Figure 4 vaccines-09-00419-f004:**
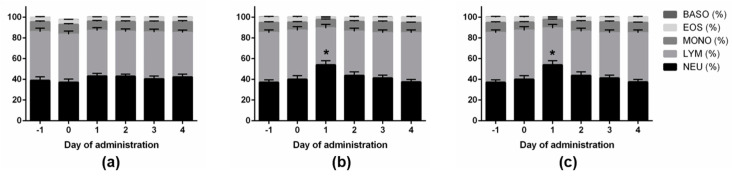
The percentages of leucocytes among the treatment groups (*n* = 60). (**a**) The control group was administered an i.m. injection of 0.9% physiological saline (10 mL/head), (**b**) the LPS group was administered an i.m. injection of lipopolysaccharide (2.5 mg/kg of body weight), and (**c**) the FMDV vaccine group was administered an i.m. injection of foot-and-mouth disease vaccine (6 PD_50_, 2 mL/head). Neutrophils (NEUs, %), lymphocytes (LYMs, %), monocytes (MONOs, %), eosinophils (EOSs, %), basophils (BASOs, %). * Experimental group comparisons within days, *p* < 0.05.

**Figure 5 vaccines-09-00419-f005:**
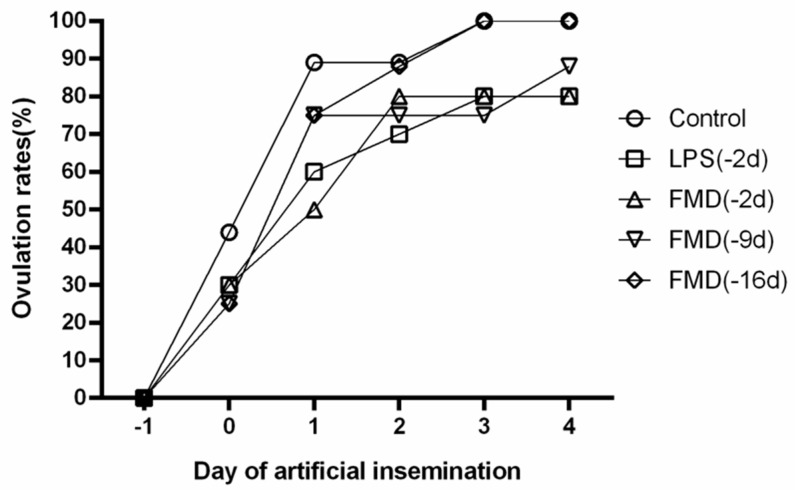
The ovulation rates and timing of the cows in the different treatments (*n* = 100). The black lines with symbols (○, □, △, ▽, ◇) represent the ovulation rate; 0 d is the time of artificial insemination. The control group (○, *n* = 20) was administered an i.m. injection of 0.9% physiological saline (10 mL/head) at −2 d, the LPS group (□, *n* = 20) was administered an i.m. injection of lipopolysaccharide (2.5 mg/kg of body weight) at −2 d, the FMDV vaccine (−2 d) group (△, *n* = 20) was administered an i.m. injection of foot-and-mouth disease vaccine (6 PD50, 2 mL/head) at −2 d, the FMDV vaccine (−9 d) group (▽, *n* = 20) was administered an i.m. injection of foot-and-mouth disease vaccine (6 PD50, 2 mL/head) at −9 d, and the FMDV vaccine (−16 d) group (◇, *n* = 20) was administered an i.m. injection of foot-and-mouth disease vaccine (6 PD50, 2 mL/head) at −16 d. Experimental group comparisons within days; *p* < 0.05.

**Figure 6 vaccines-09-00419-f006:**
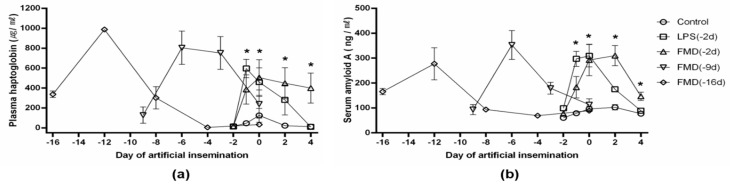
Plasma concentrations of haptoglobin and serum amyloid A (*n* = 100). The black lines with symbols (○, □, △, ▽, ◇) represent (**a**) plasma haptoglobin and (**b**) serum amyloid A levels, respectively; 0 d is the time of artificial insemination. The control group (○, *n* = 20) was administered an i.m. injection of 0.9% physiological saline (10 mL/head) at −2 d, the LPS group (□, *n* = 20) was administered an i.m. injection of lipopolysaccharide (2.5 mg/kg of body weight) at −2 d, the FMDV vaccine (−2 d) group (△, *n* = 20) was administered an i.m. injection of foot-and-mouth disease vaccine (6 PD50, 2 mL/head) at −2 d, the FMDV vaccine (−9 d) group (▽, *n* = 20) was administered an i.m. injection of foot-and-mouth disease vaccine (6 PD50, 2 mL/head) at −9 d, and the FMDV vaccine (−16 d) group (◇, *n* = 20) was administered an i.m. injection of foot-and-mouth disease virus vaccine (6 PD50, 2 mL/head) at −16 d. * Experimental group comparisons within days; *p* < 0.05.

**Table 1 vaccines-09-00419-t001:** Pregnancy rates and timing following different treatments (*n* = 100).

Group	No. of Pregnant Cows	Total	Pregnancy Rate (%)
Control	11	20	55.0
LPS (−2 d)	10	20	50.0
FMDV vaccine (−2 d)	4	20	20.0
FMDV vaccine (−9 d)	11	20	55.0
FMDV vaccine (−16 d)	12	20	60.0
Total	46	20	46.0

## Data Availability

The data presented in this study are available on request from the corresponding author.

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
