# Peer review of "Effect of Foot-and-Mouth Disease Vaccination on Acute Phase Immune Response and Anovulation in Hanwoo (Bos taurus coreanae)"

_vaccines, 2021, doi:10.3390/vaccines9050419_

Round 1
Reviewer 1 Report
In this manuscript, the authors investigate changes in acute-phase protein levels and ovulation rate when an FMDV was administered at different times before artificial insemination. It was concluded that injecting Korean cattle with an FMDV decreased the ovulation rate when administered 2 days compared with 9 or 16 days before artificial insemination. Acute-phase reaction proteins, such as haptoglobin and SAA, were increased for 8–9 days after the FMDV was administered. The pregnancy rate decreased two-fold when the FMDV was injected 2 days before artificial insemination. These results suggest that FMDV protocols should be changed to minimize the negative effects related to reproductive performance in cows. Artificial insemination should only be considered 10 days after an FMDV is administered to avoid the acute-phase immune reaction and anovulation.
General comments
In addition to LPS, the control group should also include another vaccine to conclude that acute immunity effect is specific to FMD vaccine.
Have you investigated cytokines because the article pointed out that FMDV-induced inflammation increases acute-phase proteins and delays ovulation due to inflammatory cytokines and acute-phase immune responses?
The manuscript was well written. This work is valuable. However, the manuscript needs some revision.
Author Response
Thank you for kind your comments.
Please see the attachment.

Reviewer 2 Report
The manuscript by Kim et al reports the effects of FMD vaccine on ovulation in Korean cattle when administered before artificial insemination. Authors report an increase in WBC and neutrophil count and ruminal temperature a day after vaccination. Ovulation was delayed and the rate of ovulation also dropped more when the vaccine was administered 2 day before the insemination. Compared to the control herd, plasma concentrations of haptoglobin and serum amyloid levels were higher in the herd vaccinated 2 days prior to the insemination. The pregnancy rates were also at a low of 20% is this group. However, groups that received vaccine 9 or 16 days before insemination faired better at ovulation rates. Likewise, haptoglobulin and serum amyloid concentrations recovered to normal range within 9 days. Pregnancy rates in these groups were comparable to the control.
While the data per say is interesting and clear, the results are poorly described in the main text. As a result, the reader is often left confused. The manuscript needs to be proof-read carefully for spellings, grammar, and sentence construction.
Specific comments:
Line 41 change to “plant-based”
Line 44-45 consider rephrasing the sentence for clarity.
Line 52-55 consider sentence reconstruction- “The expression of genes related to the acute-phase immune response such as steroidogenic acute regulatory protein, Toll-like receptor 4, tumor necrosis factor alpha (TNF-a), decreases significantly upon intravenous LPS injection.
Line 61 due to the “imbalance”
Figure 2. What do the authors attribute the increase in ruminal temperature between 30-36 hours to?
Figure 4. Does increase in Neutrophil count contribute to the overall increase in WBC as shown in Figure 3?
Line 144-145 “The number of WBCs increased gradually in the FMD group from days 1 to 3 compared to the control group. Figure 3 does not show an increase in the WBCs in the FMD group from days 1 to 3, although its higher than the control.
Line 149 Consider rephrasing the sentence to- “Delayed ovulation was observed in the LPS(-2d) and FMD (-2d) groups 1 day after artificial insemination, with an ovulation rate of about 60%.
Line 152-153 Rephrase as “The plasma levels of haptoglobin and SAA were greater in the LPS (−2 d) and FMD 152 (−2 d) groups 2 days after artificial insemination (= 4 days after administration), compared to the control group”.
Line 153-155 The plasma levels of haptoglobin does not seem of increase until day 4 as suggested by authors. Fig 6a shows that the levels decrease.
Line 155-157 “The plasma levels of haptoglobin and SAA were greater in the FMD (−16 d) group 8 days after administration and those in the FMD (−9 d) group were greater 9 days after administration (Figure 6).”
It is unclear how the authors arrive at this interpretation. According to Figure 6a plasma levels of haptoglobin are highest 4 days after administration in the FMD (−16 d) group and 3 days after administration in the FMD (−9 d) group. Likewise, Fig 6b Serum amyloid A levels also peaked highest 4 days after administration in the FMD (−16 d) group and 3 days after administration in the FMD (−9 d) group.
Figure 6 Legend line 205-206 “The black lines 205 with symbols (â—‹, â–¡, â–³, â–½, â—‡) represent the ovulation rates”. Change to “represents a) Plasma haptoglobin and b) Serum amyloid A levels respectively”.
Author Response

(The authors gave the same response as above.)
